# The Role of MicroRNA-Based Strategies in Optimizing Plant Biomass Composition for Bio-Based Packaging Materials

**DOI:** 10.3390/plants14182905

**Published:** 2025-09-18

**Authors:** Ayaz M. Belkozhayev, Arman Abaildayev, Bekzhan D. Kossalbayev, Aygul Kerimkulova, Danara K. Kadirshe, Gaukhar Toleutay

**Affiliations:** 1Department of Chemical and Biochemical Engineering, Geology and Oil-Gas Business Institute Named After K. Turyssov, Satbayev University, Almaty 050043, Kazakhstan; a.abaildayev@satbayev.university (A.A.); kossalbayev.bekzhan@gmail.com (B.D.K.); aigul.kerimkulova@satbayev.university (A.K.); 2Faculty of Biology and Biotechnology, Al-Farabi Kazakh National University, Al-Farabi Ave. 71, Almaty 050040, Kazakhstan; 3Ecology Research Institute, Khoja Akhmet Yassawi International Kazakh Turkish University, Turkistan 161200, Kazakhstan; 4International Faculty, Asfendiyarov Kazakh National Medical University, Almaty 050012, Kazakhstan; kadirwe.d@kaznmu.kz; 5Department of Chemistry, University of Tennessee, Knoxville, TN 37996, USA

**Keywords:** miRNAs, biomass quality, cellulose, lignin, bio-based packaging, genetic engineering, sustainable materials

## Abstract

The growing demand for sustainable alternatives to petroleum-based plastics has driven interest in bio-based packaging derived from renewable plant biomass. Cellulose, the most abundant biopolymer on Earth, provides high tensile strength, water resistance, and biodegradability, making it a key raw material for eco-friendly packaging. However, its extraction and processing are hindered by lignin, a complex polymer that adds structural rigidity but reduces cellulose accessibility. Recent research has identified plant microRNAs (miRNAs) as powerful post-transcriptional regulators capable of modifying cell wall composition by simultaneously targeting multiple genes involved in lignin biosynthesis, cellulose synthesis, and secondary cell wall formation. By fine-tuning specific miRNAs, it is possible to increase cellulose yield, reduce lignin content, and enhance overall biomass productivity without severely compromising plant growth or stress tolerance. This review summarizes the roles of major plant miRNAs in biomass regulation and outlines biotechnological strategies such as transgenic overexpression, target mimicry, artificial miRNAs (amiRNAs), and CRISPR-based editing for improving bio-based packaging feedstocks. Harnessing miRNA-mediated gene regulation offers a promising pathway toward producing high-quality biomass with optimized cellulose–lignin ratios, enabling more efficient, cost-effective, and sustainable packaging material production.

## 1. Introduction

The growing demand for renewable, eco-friendly raw materials is placing plant biomass at the forefront of sustainable material production. The quality and yield of biomass are critical parameters that determine its efficiency and cost-effectiveness in industrial applications. Biopolymers such as plant-derived cellulose and bacterial cellulose serve as key raw materials for bio-based packaging production, as cellulose is the most abundant biopolymer on Earth and plays an important role in the development of new materials [1]. Biotechnological modification of plant-derived cellulose using specific microorganisms and bacteria makes it possible to improve its structural and functional properties. As a result of such processing, bacterial cellulose with high mechanical strength, impermeability to gases and liquids, and biodegradability is obtained, making it an effective raw material for producing eco-friendly packaging. However, the qualitative composition of plant biomass, particularly the content of cellulose, hemicellulose, and lignin, directly affects the efficiency of the packaging material production process [2,3]. For example, a high lignin content makes cellulose extraction and processing more difficult, thereby increasing production costs [4]. Therefore, enhancing the biomass yield of crops grown for packaging purposes and optimizing their composition has become an important objective. In this context, special attention is being given to microRNA (miRNA) molecules. miRNAs are small RNAs, approximately 21 nucleotides in length, encoded in the plant genome, that regulate gene expression at the post-transcriptional level by binding to complementary mRNAs [5,6]. In plants, miRNAs participate in numerous biological processes, including the regulation of growth, developmental stages, hormonal and metabolic pathways, and stress responses [7]. Recent studies have revealed that plant miRNAs influence cell wall biosynthesis, including the formation of lignin and cellulose [8]. Therefore, targeted modification of miRNAs can improve plant biomass composition and increase productivity. In addition to their role in biomass composition, miRNAs are also key mediators in plant responses to abiotic stresses such as salinity, drought, and temperature fluctuations, as well as to biotic stresses like pathogens and herbivores [9,10]. By fine-tuning gene expression networks, they help plants balance growth and defense, which is crucial for sustaining high yields under challenging environmental conditions. With advances in genetic engineering tools such as CRISPR/Cas systems, artificial miRNAs (amiRNAs), and target mimicry, it is now possible to precisely manipulate specific miRNA pathways to achieve desired traits in bioenergy and fiber crops [11,12]. These strategies not only enhance biomass yield and quality but also contribute to the development of resilient crops suitable for sustainable bio-based packaging production [13,14]. This review article analyzes the role of miRNAs in enhancing plant biomass quality and productivity, as well as their potential applications in bio-based packaging production.

## 2. Main Components of Plant Biomass and Their Impact on Packaging

The structure of plant biomass is primarily composed of three main polymers: cellulose, hemicellulose, and lignin [15]. Cellulose is a linear polysaccharide made up of glucose units and forms the main framework of the plant cell wall. Cellulose fibers possess high mechanical strength, water resistance, and biodegradability, making them a fundamental raw material for packaging products such as paper and bioplastics [16]. Although cellulose is the cheapest and most widely available polymer, obtaining it in pure form is not easy; its extraction from plant tissues requires pretreatment processes such as pulping, chemical hydrolysis, or enzymatic hydrolysis [17].

Hemicelluloses are a group of branched, amorphous polysaccharides such as xylans, mannans, and glucomannans [18]. They fill the spaces between cellulose microfibrils, providing flexibility to the plant cell wall. During pulp processing, hemicelluloses are partly degraded, but some remain in the fibers and can improve paper strength by promoting bonding between fibers. Therefore, the hemicellulose content plays an important role in determining the mechanical and physical characteristics of packaging materials [19,20].

Lignin is a complex, three-dimensional aromatic polymer formed from phenylpropanoid monomers. It provides plants with strength, water impermeability, and resistance to pests and pathogens. However, from the perspective of packaging production, lignin is considered one of the most problematic components [21]. A high lignin content in raw materials requires large amounts of chemical reagents and energy during cellulose extraction. In addition, paper products containing lignin residues tend to yellow and become brittle over time [22]. Therefore, in cellulose production, efforts are made to remove lignin from the feedstock as much as possible or to reduce its content directly in the plant. Reducing lignin levels or modifying its structure is one of the main goals for facilitating biomass breakdown into sugars, producing bioethanol, and improving paper and textile manufacturing processes [23,24].

The quality of plant biomass for packaging is characterized by a high content of cellulose and hemicellulose and a relatively low lignin content. For example, in certain wood species, reducing the proportion of lignin has been shown to increase cellulose yield and improve the efficiency of the pulping process [25,26]. For this reason, researchers are exploring genetic modification strategies to alter the composition of plant cell walls [27]. For example, Yu et al. developed a CRISPR–Cas9-based gene-editing/complementation strategy to knock out a lignin biosynthesis gene in *Arabidopsis* while simultaneously expressing a modified version of the same gene in a tissue-specific manner. This approach reduced lignin content without compromising biomass yield and resulted in up to a fourfold increase in cell wall sugar yield per plant, with the phenotype remaining stable over at least four generations [28]. Recently, attention has shifted to plant endogenous regulators, particularly miRNAs, as a means of indirectly suppressing lignin biosynthesis while enhancing cellulose formation and overall biomass accumulation.

## 3. Functions of miRNAs in Plants—Biogenesis, Gene Regulation, and Cell Wall Formation

In plants, miRNA biogenesis is a multi-step process that begins with the transcription of miRNA genes by RNA polymerase II into primary miRNAs (pri-miRNAs), which fold into stem–loop structures [29]. These are processed by DICER-LIKE 1 (DCL1, an RNase III endonuclease) together with cofactors such as SERRATE (SE, zinc finger protein) and HYPONASTIC LEAVES 1 (HYL1, double-stranded RNA-binding protein) to produce precursor miRNAs (pre-miRNAs). The pre-miRNAs are then methylated by HUA ENHANCER 1 (HEN1, methyltransferase) and exported from the nucleus by HASTY (HST1, exportin-5 ortholog). In the cytoplasm, further processing yields mature miRNAs that are incorporated into the RNA-induced silencing complex (RISC), guiding ARGONAUTE 1 (AGO1) to complementary mRNAs for cleavage or translational repression (Figure 1). This tightly regulated pathway plays a vital role in post-transcriptional gene regulation, stress responses, and plant adaptability [7,30,31].

In plants, miRNAs regulate numerous processes, including the development of leaves and stems, the formation of trichomes and roots, the timing of flowering, and meristem activity [32]. They also influence responses to abiotic and biotic stresses and modulate phytohormone signaling. A key feature of miRNAs is their ability to regulate multiple genes simultaneously, which allows them to induce pleiotropic changes in the plant phenotype [33,34,35].

Regarding cell wall composition and biomass accumulation, several conserved miRNA families play a crucial role (Figure 2). They can indirectly regulate cellulose and lignin synthesis by controlling enzymes of the phenylpropanoid pathway, transcription factors, or hormone signaling [36,37]. For example, miR156 in many plants suppresses the expression of SQUAMOSA Promoter-Binding Protein-Like (SPL) transcription factors. Since SPL factors regulate developmental transitions, including flowering, branching, and secondary growth, miR156 can indirectly influence biomass production and cell wall properties [38]. Another example is miR397 and its related miRNAs, miR408 and miR857, which directly target the genes of multicopper laccase enzymes involved in lignin polymerization and degrade their transcripts [39]. As a result, lignin biosynthesis slows down, and the proportion of lignin in the plant cell wall decreases. Similarly, miR319 suppresses TCP-type transcription factors, altering the growth morphology of leaves and stems and influencing genes involved in secondary cell wall formation. Overall, plant miRNAs are key nodes in gene regulatory networks. By redirecting their activity, it is possible to simultaneously influence multiple genetic pathways and coordinately modify complex phenotypic traits, such as biomass yield and the cellulose-to-lignin ratio. This capability makes miRNAs valuable tools in plant breeding and biotechnology, particularly for producing plant raw materials suitable for bio-based packaging.

## 4. Enhancing Plant Biomass Quality and Yield Through miRNAs

Recent studies have shown that genetic modification of certain plant miRNAs can significantly improve both the quality and quantity of plant biomass [40] (Table 1). For example, miR156 is highly expressed during the juvenile phase of plant development and decreases with age. miR156 binds to the mRNAs of numerous SPL transcription factors, suppressing their expression. As a result, it prolongs the vegetative growth phase, delays flowering, and increases branching and leaf number [41,42]. For example, Gao et al. performed a transcriptome analysis to investigate the molecular mechanisms of miR156 overexpression in *Medicago sativa*. Using RNA-seq on two transgenic lines (A11a and A17), they found significant changes in the expression of multiple genes, with miR156 markedly downregulating several SPL transcription factors (MsSPL6, MsSPL12, MsSPL13, MsSPL2, MsSPL3, MsSPL4, MsSPL9). These genes are involved in starch and sucrose metabolism, flavonoid biosynthesis, and lignin catabolism. The study demonstrated that miR156 overexpression not only enhances biomass yield in alfalfa but also alters cell wall composition [43]. These transgenic alfalfa plants exhibited delayed flowering and shorter stature but produced more stems and leaves, resulting in improved overall forage quality. Such traits could be directly beneficial for the bio-based packaging industry, as higher biomass with optimized lignin content provides more efficient raw material for paperboard, molded fiber, and nanocellulose production. Similarly, in miscanthus and maize, the *de-etiolated1* (Corngrass1) mutation constitutively increases miR156 levels, causing the plants to remain in the juvenile phase, which results in shorter stems and increased tillering [44]. Conversely, reducing miR156 activity either by disrupting MIR156 genes through CRISPR or by sequestering it with a target mimic induces early adult traits in plants, leading to accelerated flowering and stem elongation [45,46]. This approach has been investigated in wheat, where MIM156 lines designed to inhibit miR156 developed larger leaves and longer roots, resulting in higher initial biomass and the ability to allocate more resources toward spike formation. Thus, fine-tuning the miR156/SPL module can reshape plant architecture, enhancing both biomass accumulation and its subsequent conversion into products such as grain or fiber [47]. In a packaging context, such fine-tuning could ensure a steady and scalable supply of cellulose-rich biomass, reducing costs for industrial processors and improving material properties of biodegradable packaging.

A similar strategy applies to other miRNAs, such as miR319, which targets the mRNAs of TCP transcription factors controlling leaf shape, stem elongation, and secondary thickening. For example, Liu et al. demonstrated that overexpressing Osa-miR319b in switchgrass increased plant height and biomass yield, reduced lignin content, and enhanced enzymatic hydrolysis efficiency, whereas suppression of miR319 (MIM319) had the opposite effects. The miR319–PvPCF5 module was identified as a key regulator of biomass quality [48]. Lower lignin and higher cellulose availability in such crops can improve pulping efficiency and reduce chemical use in producing recyclable paper and nanocellulose-based coatings for packaging applications. Additionally, to boost crop productivity, researchers have developed transgenic lines that modulate miRNA expression. In tomato, Ori et al. showed that overexpressing miR319 enlarges leaflets and promotes continuous growth of the leaf margin, markedly altering leaf morphology [49]. miR396 is a highly conserved plant miRNA that controls growth by targeting growth-regulating factors (GRFs) and modulating gene expression post-transcriptionally. In *Arabidopsis* and rice, altered levels of miR396a and miR396b reduce the expression of six GRF transcription factors, leading to narrower leaves, abnormal cell growth, and excessive cell division [50,51].

Patel et al. reported that overexpression of native Musa-miR397 in banana plants enhanced biomass production by 2–3 fold compared to wild type, without compromising tolerance to copper deficiency or NaCl stress. The upregulation of miR397 significantly downregulated laccase genes, reducing lignin biosynthesis while maintaining stress resilience, highlighting miR397 as a promising target for improving biomass quality and yield [39]. For the packaging sector, miR397’s ability to lower lignin while maintaining stem strength can yield feedstocks better suited for high-strength paper products and molded fiber containers. miR408 is functionally similar to miR397, also regulating copper-containing proteins such as laccases and plastocyanin-like proteins, and is highly conserved in plants. A recent study in white poplar showed that overexpression of Pag-miR408 significantly increased cell wall degradability, eliminating the need for acid pretreatment to release sugars from biomass [52]. High miR408 expression in poplar promoted faster growth and larger leaves, while its knockout reduced cellulose accessibility and stunted growth, demonstrating that miRNAs can help balance biomass quality with growth rate. Such characteristics are highly relevant for producing raw materials for sustainable packaging, where easy fiber processing and high yield are critical for reducing environmental impact and production costs.

**Table 1 plants-14-02905-t001:** Plant miRNAs and their effects on biomass composition and yield.

miRNA	Primary Target Genes/Pathways	Impact on Biomass	References
miR156	SPL transcription factors (e.g., *SPL3*, *SPL4*, *SPL9*)	Extends the juvenile phase, increases branching, and delays flowering. Moderate overexpression enhances biomass yield, increases cellulose content, and reduces lignin. Excessive expression can cause dwarfism and delayed flowering.	[38,43,53]
miR319	TCP transcription factors (e.g., *PvPCF5*)	Suppresses TCP genes controlling leaf and stem growth, increasing stem length and plant height. Overexpression reduces lignin biosynthesis, lowers lignin percentage, and improves saccharification efficiency.	[48]
miR397	Laccase enzymes (*LAC* genes)	Directly represses laccase genes required for lignin polymerization. Overexpression decreases lignin content by 15–20%, increases relative cellulose proportion, and improves processing efficiency, though it may slightly reduce mechanical strength.	[4]
miR408	Laccases, plastocyanin (Cu-dependent proteins)	Delays secondary growth and lignification, increasing cell wall accessibility. Overexpression in poplar enhances saccharification without pretreatment and accelerates plant growth while slightly modifying the S/G lignin ratio. Also modulates copper–protein balance under stress.	[52]
miR828/miR858	MYB transcription factors (phenylpropanoid pathway)	Repress MYB factors regulating phenylpropanoid metabolism, altering lignin–anthocyanin balance. In *Arabidopsis*, the miR858a–MYB module coordinates lignin and flavonoid biosynthesis; in *Populus*, miR828 reduces lignin by targeting lignin-specific MYBs. Direct effects on biomass yield remain less studied but show potential for wood quality improvement.	[54,55]
miR396	Growth-regulating factors (GRFs) and GRF-interacting factors	Knock-out of MIR396e/f increases grain size, panicle branching, and above-ground biomass, especially under nitrogen limitation.	[56]
miR160	Auxin response factors (*ARF10/16/17*)	Regulates auxin signaling; modulation alters root and shoot development, affecting biomass partitioning and accumulation.	[57,58]
miR167	Auxin response factors *ARF6/ARF8*	Controls lateral root growth and fertility; altering miR167 expression modulates root biomass and influences overall biomass yield.	[57,59]
miR164	NAC domain transcription factors (e.g., *CUC* genes)	Involved in lateral root formation and leaf senescence; altering its level influences root biomass and senescence timing.	[57,60]
miR159	MYB transcription factors	Regulates the transition from juvenile to adult phase and anther development; altering miR159 can modify flowering time and biomass allocation.	[57,61]
miR171	GRAS transcription factors (e.g., *SCL6*)	Controls axillary meristem development and shoot branching; influences tiller number and vegetative biomass.	[57,60]
miR172	APETALA2 (AP2)-like transcription factors	Promotes phase transition and flowering; altering miR172 influences biomass by shifting resources from vegetative growth to reproductive organs.	[57,62]
miR393	Auxin receptors (*TIR1*, *AFB2*, *AFB3*)	Suppression of auxin receptor genes reduces auxin sensitivity, modifies branching and root architecture, affecting biomass distribution.	[63]
miR444	MADS-box transcription factors in monocots	Regulates tillering and root development in rice; overexpression affects nitrogen uptake and biomass accumulation.	[64]
miR528	Laccases and auxin-responsive factors	In monocots (rice/maize), miR528 regulates lignin biosynthesis and stress responses; altering its expression can reduce lignin and improve biomass digestibility.	[65,66]
miR399	PHO2/UBC24 ubiquitin conjugase	Controls phosphate homeostasis; miR399 overexpression can increase shoot biomass under phosphate limitation by improving phosphate allocation.	[67,68]
miR827	NLA (nitrogen limitation adaptation) gene	Regulates nitrogen remobilization and phosphate transport; manipulation influences biomass yield under nutrient limitation.	[69]
miR398	Cu/Zn superoxide dismutases (*CSD1/2*), copper chaperones	Regulates reactive oxygen species homeostasis and copper distribution; modulation affects stress tolerance and indirectly alters biomass accumulation.	[70]
miR395	ATP sulfurylases (*APS*), sulfate transporters	Controls sulfate assimilation; overexpression can improve sulfur use efficiency and biomass production in sulfur-limited conditions.	[71,72]
miR535	SPL family genes	Modulate panicle branching and grain size in rice and sorghum; altering their levels affects biomass distribution between vegetative and reproductive organs.	[73,74]

## 5. Biotechnological Approaches Using miRNAs to Enhance Plant Biomass Quality and Yield

### 5.1. In Silico Analysis of miRNA Target Genes

Before initiating experimental work on the use of miRNAs for plant improvement, bioinformatics tools are first employed to identify miRNA molecules that interact with the target genes. At this stage, regulatory gene networks are analyzed, and targets involved in cellulose and lignin biosynthesis are selected [75]. Currently, a number of advanced web-based platforms for predicting miRNA–mRNA interactions in plants are widely used. Examples include TargetFinder, psRNATarget, miRTarBase, RNAhybrid, TAPIR, and MirTarget [7,76,77] (Table 2). TargetFinder predicts plant miRNA–mRNA duplexes with high accuracy by employing position-based weighted scoring that accounts for mismatches and gaps [78,79]. psRNATarget forecasts the binding of plant miRNAs to their target mRNAs based on sequence complementarity and target site accessibility (UPE), allowing parameter customization to identify both canonical and non-canonical targets, and supports the analysis of large transcript libraries [80]. miRTarBase is a comprehensive repository of experimentally validated miRNA–target interactions (MTIs); by integrating data on expression profiles, tissue specificity, and sequence variants, it aids in elucidating miRNA regulation under stress conditions and in constructing gene network models [81,82]. RNAhybrid specializes in identifying duplexes with the lowest free energy (ΔG), thereby pinpointing critical binding regions [83,84]. TAPIR combines two algorithms: a FASTA-based rapid search and precise RNAhybrid analysis to detect “imperfect” duplexes that might otherwise be overlooked; by tuning parameters such as ΔG and mismatch tolerance, it enables the discovery of complex regulatory phenomena, including target mimicry [85,86]. MirTarget predicts potential binding sites within 5′UTR, CDS, and 3′UTR regions, accounting for non-canonical base pairs such as G–U and A–C [87,88,89,90].

### 5.2. Biotechnological Strategies for Applying miRNAs

Several biotechnological strategies are employed to enhance plant traits, including transgenic overexpression, short tandem target mimic (STTM), amiRNAs, and genome editing. These approaches can be used to increase cellulose yield and optimize lignin content—two priorities for bio-based packaging production (Figure 3) [91].

Transgenic overexpression and RNA interference involve introducing the precursor of the MIR gene of interest into the plant under the control of a strong promoter. As a result, the level of that miRNA increases significantly, leading to reduced expression of its target genes [91,92]. Conversely, when the aim is to reduce the activity of a specific miRNA, an artificial RNA molecule is expressed that serves as its complementary decoy. This molecule sequesters the miRNA, preventing it from binding to its target mRNA; this approach is known as target mimicry. STTM is a small transcript containing two miRNA-binding sites separated by a 3–4 nucleotide spacer, which is stably expressed in plant cells and effectively suppresses the function of the corresponding miRNA [93,94]. RNAi technology can also be used to silence miRNA precursors or their target genes through double-stranded RNA, but it is now often replaced by more precise approaches such as amiRNA or clustered regularly interspaced short palindromic repeats (CRISPR) [95]. Peng et al. developed a large-scale resource of STTMs targeting both conserved and species-specific miRNAs in *Arabidopsis*, tomato, rice, and maize. They demonstrated that tissue-specific inactivation of certain miRNAs in rice, such as via STTM156/157 and STTM165/166, increased grain size without negatively affecting overall plant growth and development. Transcriptomic and metabolic analyses revealed that these STTMs modulated pathways related to plant hormone biosynthesis, secondary metabolism, and ion-channel activity. This study confirmed that STTM is an effective and versatile approach for dissecting miRNA functions and improving crop traits [96].

In the amiRNA technology, a 21-nucleotide sequence complementary to the target gene’s mRNA is inserted into the precursor of the plant’s own miRNA. In other words, the natural miRNA scaffold is used to design a new miRNA that specifically silences the desired gene. Unlike traditional RNAi, amiRNA offers very high specificity with minimal off-target effects [97]. For example, if it is necessary to precisely suppress a particular lignin biosynthesis gene that hinders paper production, an amiRNA complementary to the transcript region of that gene can be designed and introduced into the plant [98]. For example, Shafrin et al. showed that amiRNA targeting C3H and F5H genes in jute reduced stem lignin by ~25% and fiber lignin by 12–15%, demonstrating its potential to improve fiber quality in industrial crops [99]. The amiRNA method has been successfully applied in rice, tobacco, and other species to study gene functions. In the future, this approach also holds potential for multiplex formats to regulate multiple genes simultaneously [100]. Warthmann et al. demonstrated that amiRNAs in rice can specifically silence target genes with stable inheritance, proving their effectiveness for functional genomics and crop improvement in monocots [101].

The CRISPR/Cas9 system has emerged as an innovative tool in the field of miRNA research [102]. Using CRISPR to knock out the MIR gene itself, if a particular miRNA has a negative impact on the plant, involves creating small deletions or mutations in the gene to abolish its function. In addition, modifying the 3′UTR region of a miRNA target by introducing point mutations in the miRNA recognition site can render the mRNA invisible to that miRNA [103,104]. Currently, studies using CRISPR/Cas9 have successfully knocked out lignin biosynthesis genes in trees, achieving up to a 50% reduction in lignin content [105]. Recent studies show that Li et al. demonstrated lignin biosynthesis can be suppressed while simultaneously enhancing cellulose synthesis in *Populus*. By overexpressing a mutated transcriptional repressor (*PdLTF1^AA*) to reduce lignin and introducing cellulose synthase genes (*PdCesA4*, *PdCesA7A*, *PdCesA8A*) to boost cellulose production, the transgenic plants exhibited significantly lower lignin content, higher cellulose content, and larger xylem fiber cell diameters. This dual-target approach provides a promising strategy for improving fiber quality and biomass processing efficiency [106].

Another promising miRNA-based approach is their integration into classical breeding programs. If a particular miRNA or allelic variation in its target is found to affect yield, that allele can be propagated through conventional hybridization and selection. For example, in rice, a morphant allele of the OsSPL14 gene was found to promote increased tillering and yield and was used to develop an ideotype variety for the new green revolution [107]. In maize, the Corngrass1 mutation (excess miR156) is considered an undesirable trait because it reduces yield, so breeders aim to eliminate it. In genomic selection, SNP variants in MIR genes or variations in their target genes can be used as DNA markers. For instance, if certain natural tree populations carry weak miRNA-binding mutations in the 3′UTR of LACCASE genes, this could result in a low-lignin phenotype, making such trees of practical interest for selection using molecular markers [108,109].

In the future, efforts are expected to go beyond modifying a single miRNA, aiming instead to alter multiple regulatory elements simultaneously. For example, increasing the expression of both miR156 and miR397 in the same plant could synergistically improve biomass quantity and cellulose quality. Such multigene modifications can be achieved using CRISPR or by simultaneously altering transcription factors [110,111]. Excessive reduction of lignin content can soften plant wood, leading to lodging; however, if miRNA expression is enhanced only in cambium and xylem cells, the effect will be limited to the wood while other organs develop normally. To achieve such precision, synthetic promoters and inducible expression systems are being explored [112]. Table 3 summarizes the main advantages and disadvantages of each biotechnological strategy discussed above, based on their application to optimizing cellulose–lignin ratios for bio-based packaging materials. Overall, targeting miRNAs at the molecular and genomic levels has opened new opportunities in plant breeding and bioengineering. Traits that typically require decades to improve through conventional breeding, such as biomass yield and wood quality, can now be restored or enhanced within just a few generations using miRNA-based approaches.

## 6. Prospects of Plant miRNA Research in Bio-Based Packaging Production

The development of bio-based packaging in the coming decades may largely depend on the availability and quality of plant biomass. In the consumer market, there is a growing demand for cellulose-, starch-based, and fully biodegradable materials as environmentally friendly alternatives to petroleum-derived plastics, which are known for their ecological harm [128]. To meet these requirements, it is necessary to develop high-yield plant varieties rich in cellulose and suitable for processing. miRNA-based biotechnology is one of the promising approaches for producing such varieties. miRNAs can comprehensively modify the composition of the cell wall. While traditional methods often produce limited effects, miRNAs can simultaneously regulate the expression of multiple enzyme genes, enabling a balanced adjustment of the cellulose–lignin–hemicellulose ratio [8,129]. It is also effective for improving the properties required for this packaging. For example, the cell-wall-related miR775 has been shown to regulate leaf cell wall properties by modulating pectin levels and the elastic modulus of plant tissues, demonstrating miRNA’s potential to alter wall mechanics in a way that could be transferred to biomass suitable for packaging materials [130]. In parallel, nanocellulose derived from high-cellulose biomass is increasingly applied in food packaging due to its excellent mechanical strength, barrier properties, and biodegradability. Industrial production of nanocellulose for packaging has progressed from lab-scale to pilot and commercial levels across Europe, Africa, and Asia [131]. Moreover, Virginia Tech researchers recently developed a low-pressure treatment that strengthens cellulose nanofibril films with significantly less energy consumption, improving gas barrier properties, mechanical strength, and transparency, key attributes for packaging applications [132]. Real-world applications of such cellulose-based innovations are already visible in the packaging industry. For instance, Huhtamaki’s ICON^®^ paper-based ice cream containers (95% renewable bio-based fiber) [133] and UPM Specialty Papers’ compostable, bio-based coated food packaging [134] demonstrate how high-quality cellulose feedstocks can be transformed into market-ready, recyclable, and compostable products.

Integrating miRNA-engineered high-cellulose crops into these supply chains could reduce production costs, enhance material performance, and lower the carbon footprint of large-scale packaging manufacturing. Together, these findings suggest a powerful synergy in which miRNA engineering could optimize plant biomass for higher cellulose content and improved mechanical properties, while industrial processing, such as nanocellulose production and low-pressure strengthening treatments, can convert this biomass into high-performance biodegradable packaging. In addition, miRNAs can exert their effects while maintaining the plant’s overall growth rate and stress tolerance [135]. Excessive suppression of lignin can weaken stems and slow growth, but miRNAs offer fine-tuned regulation, reducing levels without complete shutdown. For example, miR408 slightly lowers lignin while boosting growth, and optimal miR156 increases biomass, though too much slows growth [38,52]. Similarly, Zhang et al. demonstrated that the miR397a–LAC17 regulatory module in *Medicago ruthenica* modulates lignin biosynthesis by targeting the laccase gene *MrLAC17*. Reduced miR397a abundance increased *MrLAC17* expression and lignin content, enhancing stem strength, whereas restoring *MrLAC17* in *Arabidopsis* double mutants confirmed its role in lignin polymerization. This work highlights a potential strategy to fine-tune lignin composition for tailored biomass applications [136]. Future work should precisely control miRNA doses and timing using advanced promoters and regulation systems.

Research on plant miRNAs is unlocking new genetic resources. Although thousands of miRNAs are currently known, the functions of many of them remain largely unexplored [72,137]. It is possible that some of them have a strong influence on biomass accumulation or the synthesis of specific polymers. Sharma et al. showed that light-regulated miR858a targets R2R3-MYB factors; overexpression of miR858a suppresses these MYBs, whereas a target mimic (MIM858) increases MYB activity and redirects phenylpropanoid flux toward flavonoids at the expense of lignin, providing a lever to tune lignin content for processing-friendly biomass [54]. For example, the recently identified miRNA miR6443 was found to increase the proportion of syringyl (S) lignin monomers in poplar wood, thereby altering the overall properties of lignin [138]. Similarly, Shen et al. identified 25 conserved and 173 novel miRNAs in the developing xylem of *Pinus massoniana*, many targeting secondary cell wall biosynthesis genes (including MYB, ARF, and LAC), which are directly involved in lignin and cellulose formation [139]. By integrating such miRNA-mediated biomass tuning (optimized cellulose and lignin traits) with industrial-level nanocellulose processes, researchers can more effectively develop sustainable packaging materials that combine functional performance with eco-friendliness. In the future, scientists plan to use various omics technologies to identify new miRNA networks that influence biomass quality. Modifying the components of these networks through genome editing will serve as a foundation for developing crops that support the future bio-based economy [8]. Integrating miRNA-based innovations with breeding can translate lab results into practice by combining genetic engineering and classical methods. In conclusion, research on plant miRNAs holds great promise for improving biomass for the packaging industry.

## 7. Challenges and Limitations of miRNA-Based Biomass Improvement

One of the major challenges is the exogenous delivery of miRNA molecules into plant cells and ensuring their stable expression. For example, attempts to introduce miRNAs or miRNA-based constructs into plants often encounter the barrier of the cell wall, which can cause tissue damage and result in low efficiency. In addition, these small RNAs are prone to rapid degradation under high temperatures and unfavorable pH conditions, preventing them from reaching their targets [140,141]. The need for crop-specific genetic transformation protocols also hinders the broad application of miRNA-based approaches across all plant species. To address such challenges, researchers are exploring the use of nanoparticles as carriers. Nanotechnology has been shown to enable the efficient delivery of miRNA molecules into plants without causing damage [140,142]. The biomass improvement results achieved through miRNA-based approaches can vary depending on the plant species and even its tissue-specific characteristics. In different species, a single conserved miRNA can target different genes and display varying expression levels, making its effects highly context-dependent [143]. Therefore, an approach that is effective in one species may have reduced or negligible effects in another.

Artificially altering miRNA levels can lead to many unforeseen side effects, as each plant miRNA typically recognizes and suppresses the expression of dozens of different mRNAs. As a result, unintended genes may also be indirectly silenced, leading to unplanned phenotypic alterations in the plant. For example, it has been shown that each miRNA regulates not just one but multiple genes, and conversely, each gene can be influenced by numerous miRNAs; this interconnected network means that enhancing a particular miRNA can also trigger changes in secondary pathways [144,145]. In addition, when using genome editing methods or artificial target mimicry technology to stably alter miRNA levels, issues of genomic stability may arise. For instance, employing the CRISPR/Cas9 system carries the risk of mutations at non-target DNA sites, while random integration of transgenic constructs into the genome can disrupt essential genes or alter the expression background. Therefore, to ensure that miRNA-based modulation is restricted to the intended target genes, careful bioinformatic design, the use of specific promoters, and comprehensive experimental validation are considered essential [146,147].

Improving plant biomass quality typically involves reducing lignin content and increasing the proportion of cellulose; however, when modifying these major cell wall polymers, it is crucial not to disrupt the overall physiological balance of the plant. Lignin provides mechanical strength and serves a protective function; excessive reduction can weaken stems, slow growth, and lead to yield decline. Indeed, strong suppression of lignin biosynthesis genes has often been shown to hinder plant development and reduce overall stress tolerance [148,149]. For example, in *Arabidopsis* and *Medicago*, downregulation of the HCT gene drastically reduced lignin levels and led to excessive accumulation of the defensive phytohormone salicylic acid (SA); as a result, plant growth was inhibited, while resistance to pathogens was enhanced [150]. This study showed that excessive reduction of lignin content can drive the plant’s immune system into a hyperactive state, thereby inhibiting growth. Therefore, instead of completely shutting down lignin and cellulose synthesis, it is necessary to fine-tune and gradually regulate them to maintain the required balance. Through miRNA-based regulation, lignin levels can be slightly reduced to optimize the cellulose–lignin ratio without severely disrupting plant structure. For example, in the tea plant (*Camellia sinensis*), the CsmiR397a–CsLAC17 module was found to balance shoot lignin content, maintaining both tissue softness and resistance to the pathogenic fungus *Botrytis cinerea* (gray mold) [151].

The stability of miRNA-induced traits may depend on environmental conditions. Many plant miRNAs change expression under stresses such as drought, salinity, and temperature shifts. Thus, a miRNA modification that boosts biomass under normal conditions may be less effective under stress. Conversely, during stress, plants may upregulate certain endogenous miRNAs to limit growth and redirect resources for defense. For example, at low temperatures, alternative miRNA biogenesis pathways can activate, allowing synthesis of essential miRNAs even when some cofactors are lacking [152,153]. Such phenomena can also affect the expression of externally introduced miRNAs, causing their levels to fluctuate across environments. Moreover, adverse external factors may counterbalance or even completely negate the beneficial phenotypes obtained through miRNA modification. For instance, plants with reduced lignin levels may be more easily damaged under strong wind or pathogen pressure due to the diminished protective role of lignin. Therefore, miRNA-modified plants should be tested under various stress conditions to ensure that the acquired traits remain stable over the long term [154]. In summary, while miRNA-based strategies offer promising tools for optimizing cellulose–lignin ratios, their application must carefully balance improvements in cell wall composition with maintaining plant physiological integrity. Excessive lignin reduction, unintended gene silencing, environmental variability, and genomic stability issues remain significant risks, requiring precise regulation, context-specific design, and multi-condition testing to ensure long-term stability and agronomic performance [155,156].

## 8. Conclusions and Future Perspectives

Plant miRNAs are modern molecular tools for enhancing biomass quality and productivity, creating new opportunities for the bio-based packaging industry. By precisely regulating the expression of genes involved in cell wall biosynthesis, miRNAs can increase cellulose content, optimize lignin composition, and balance hemicellulose levels. These changes improve the mechanical strength, water resistance, and processing efficiency of biomass while reducing the need for excessive energy and chemical inputs. The key advantage of miRNA-based approaches lies in their ability to fine-tune multiple genetic pathways simultaneously. This enables targeted modifications requiring decades in conventional breeding to be achieved within just a few generations. Moreover, adjusting miRNA levels can enhance desirable traits without compromising overall plant growth, stress tolerance, or adaptability. Current biotechnological methods, such as amiRNAs, target mimicry technology, and genome editing, allow stable and highly precise trait modification. Tissue-specific promoters and inducible systems further control the timing and location of these changes, minimizing potential side effects. In the future, integrating miRNA biotechnology with traditional breeding and genomic selection can generate high-yield crops with increased cellulose and optimized lignin content. This will help reduce dependence on petroleum-based plastics, lower the carbon footprint, and accelerate the shift toward a sustainable circular bioeconomy. Furthermore, the integration of synthetic biology, multi-omics analyses, and machine learning-based genetic modeling will provide deeper insights into miRNA functions and enable targeted redesign. However, the complete spectrum of miRNA roles in plant metabolism remains to be fully understood, making comprehensive fundamental and applied research in this field essential.

## Figures and Tables

**Figure 1 plants-14-02905-f001:**
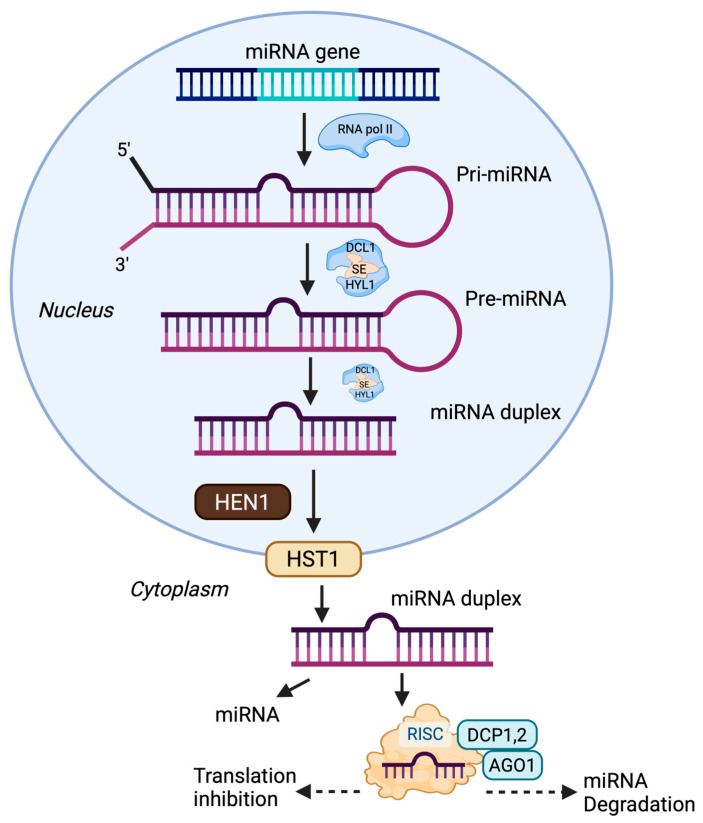
miRNA biogenesis and functional pathway in plants. miRNAs are transcribed as pri-miRNAs, processed into pre-miRNAs by DCL1 with cofactors, and further cleaved into miRNA duplexes. After methylation (HEN1) and export (HST1) to the cytoplasm, mature miRNAs are incorporated into the RISC complex to regulate target mRNAs via translation inhibition or degradation. Note: Created with BioRender, License No. OA28M9ONMN.

**Figure 2 plants-14-02905-f002:**
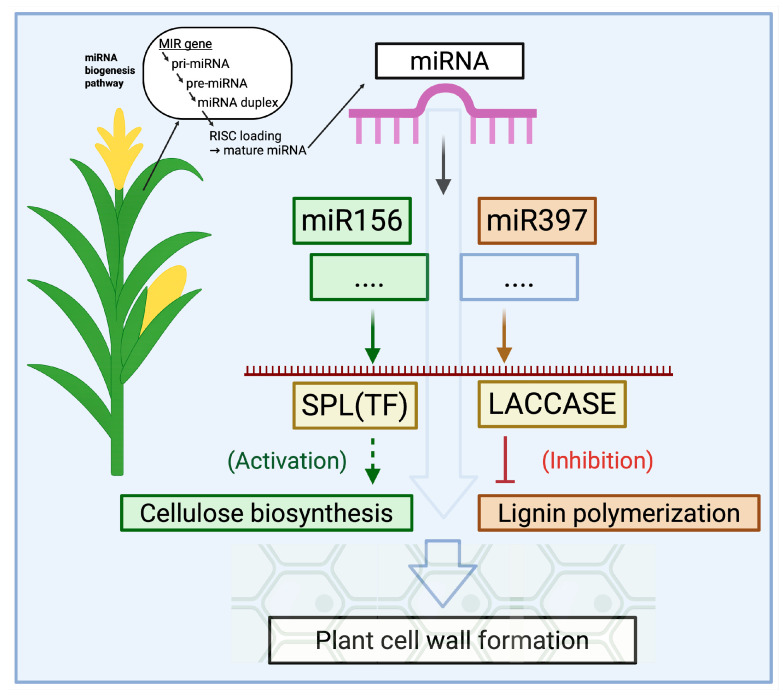
Regulation of cell wall formation genes by microRNAs in plants. Schematic representation of the miR156–SPL and miR397–LACCASE modules affecting cellulose biosynthesis and lignin polymerization, respectively. Arrows indicate activation, and T-bars indicate inhibition. The scheme illustrates an engineered regulation approach; in natural conditions, miRNA activity and their effects on cell wall composition may vary. Note: Created with BioRender, License No. VP28Q0LTJJ.

**Figure 3 plants-14-02905-f003:**
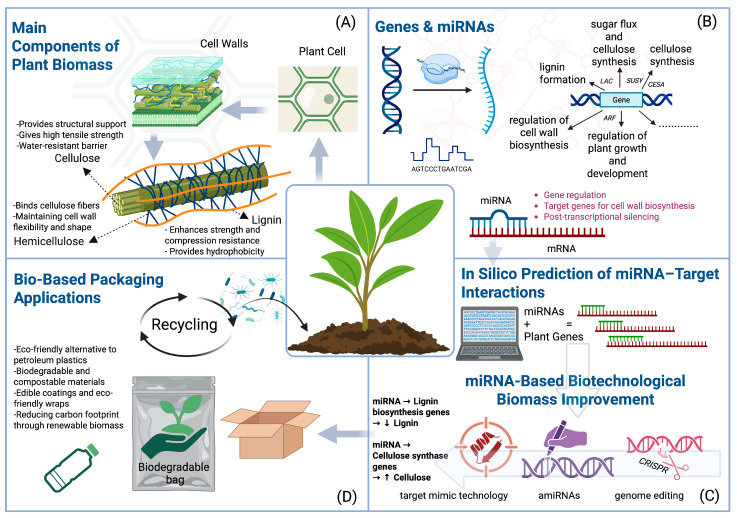
Overview of plant biomass components, gene/miRNA regulation, and biotechnological strategies for bio-based packaging applications. (**A**) Main components of plant biomass: cellulose (structural support, tensile strength, water resistance), hemicellulose (network binding, flexibility), and lignin (strength, compression resistance, hydrophobicity); (**B**) genes and miRNAs regulating lignin formation, cellulose synthesis, cell wall biosynthesis, and plant growth; (**C**) miRNA-based biotechnological approaches: target mimic technology, amiRNAs, and genome editing to decrease lignin and increase cellulose; (**D**) bio-based packaging applications: biodegradable and compostable materials, edible coatings, and recycling for reduced carbon footprint. Note: Created with BioRender, License No. TI28MORFJ3.

**Table 2 plants-14-02905-t002:** In silico prediction tools for plant miRNA–mRNA interactions.

Tool Name	Main Function/Description	URL
TargetFinder	Predicts plant miRNA–mRNA duplexes with high accuracy using position-based weighted scoring, accounting for mismatches and gaps [78,79].	http://targetfinder.org (accessed on 6 September 2025)
psRNATarget	Predicts binding of plant miRNAs to target mRNAs based on sequence complementarity and target site accessibility (UPE); customizable parameters; supports the analysis of large transcript libraries [80].	https://www.zhaolab.org/psRNATarget/ (accessed on 6 September 2025)
miRTarBase	Repository of experimentally validated miRNA–target interactions (MTIs); integrates expression profiles, tissue specificity, and sequence variants [81,82].	https://mirtarbase.cuhk.edu.cn (accessed on 6 September 2025)
RNAhybrid	Identifies miRNA–mRNA duplexes with the lowest free energy (ΔG), pinpointing critical binding regions [83,84].	https://bio.tools/rnahybrid (accessed on 6 September 2025)
TAPIR	Combines FASTA-based quick search and RNAhybrid analysis to detect imperfect duplexes; allows parameter tuning to find complex regulatory patterns, including target mimicry [85,86].	https://bioinformatics.psb.ugent.be/webtools/tapir/ (accessed on 6 September 2025)
MirTarget	Predicts potential binding sites within 5′UTR, CDS, and 3′UTR, considering non-canonical base pairs such as G–U and A–C [87,88,89,90].	https://doi.org/10.6026/97320630010423; https://www.bioinformation.net/012/97320630012237.pdf

**Table 3 plants-14-02905-t003:** Summary of biotechnological approaches for miRNA application in improving plant biomass composition for bio-based packaging.

Approach	Advantages	Disadvantages
Transgenic overexpression	Increases miRNA levels effectively; enables simultaneous downregulation of multiple target genes; relatively well-established method in plants [113,114].	Possible off-target effects; may cause pleiotropic traits; requires transformation protocols; potential regulatory restrictions [113,115].
Target mimicry (STTM)	Highly specific miRNA inhibition; stable suppression; tissue-specific expression possible; allows functional analysis [116].	Requires careful design to avoid unintended interactions; effect depends on expression stability; may not fully suppress highly abundant miRNAs [117].
Artificial miRNAs (amiRNAs)	Very high sequence specificity; minimal off-target effects; suitable for targeting single genes; adaptable to multiplex targeting [118,119].	Time-consuming design; requires sequence knowledge; may lose efficacy with target gene mutations; transformation dependent [120,121].
CRISPR/Cas-based genome editing	Permanent and heritable changes; can knock out MIR genes or alter target recognition sites; enables multiplex editing; avoids transgene retention in some cases [122].	Off-target genome edits possible; editing efficiency variable; regulatory and biosafety concerns; some targets difficult to edit [123].
Integration with breeding programs	No transgene introduction (when using natural alleles); suitable for long-term crop improvement; compatible with marker-assisted selection [124,125].	Limited to available natural variation; slower than direct engineering; trait expression influenced by genetic background [126,127].

## Data Availability

The data presented in this study are available on request from the corresponding author.

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
