# Peer review of "The Role of MicroRNA-Based Strategies in Optimizing Plant Biomass Composition for Bio-Based Packaging Materials"

_plants, 2025, doi:10.3390/plants14182905_

Round 1
Reviewer 1 Report
Comments and Suggestions for Authors
This review focuses on a topic with scientific potential. The detailed descriptions of the individual microRNA-based approaches and alternative solutions to the topic in question are valuable. Table 1 is valuable and useful. The images are concise and clear.
The review focuses on the topic of optimising the ratio of cellulose to lignin, which could enable the production of more efficient, cost-effective and sustainable packaging materials in the context of microRNA-based strategies.
In this context, it would be appropriate to modify the title of the manuscript, e.g. „The Role of microRNA-based strategies in optimising Plant Biomass composition for bio-based packaging materials“.
The phrase 'enhancing plant biomass quality' suggests that the quality of plant biomass is currently inadequate for any use.
In paragraph 3, I suggest listing the names of enzymes involved in miRNA biogenesis in plants. For example: SE (SERRATE, the zinc finger protein) etc.
Please rephrase the sentence (row 117-118):....by RNA polymerase II into primary miRNAs (pri-miRNAs) which fold into a stem-loop structure.
Please, include a scheme of how microRNAs regulate genes involved in the regulation of cell wall formation. How are the individual genes of this process regulated by microRNA?
For Paragraph 5.2, please summarise the advantages and disadvantages of each approach in the context of the review topic, presenting this information in table form.
It is beneficial that the authors consider the potential risks and disadvantages of individual approaches, addressing and arguing them specifically (e.g. lines 404–439).
For Paragraph 7, please summarize the challenges for miRNA-based approaches in the context of the review and poin out the potential risks in relation to: modified cell wall composition versus balanced plant.
Author Response
Reviewer 1.
Comments 1. This review focuses on a topic with scientific potential. The detailed descriptions
of the individual microRNA-based approaches and alternative solutions to the topic in question
are valuable. Table 1 is valuable and useful. The images are concise and clear.
The review focuses on the topic of optimising the ratio of cellulose to lignin, which could
enable the production of more efficient, cost-effective and sustainable packaging materials in
the context of microRNA-based strategies.
In this context, it would be appropriate to modify the title of the manuscript, e.g. “The Role of
microRNA-based strategies in optimising Plant Biomass composition for bio-based packaging
materials ”.
The phrase 'enhancing plant biomass quality' suggests that the quality of plant biomass is
currently inadequate for any use.
Response 1. We appreciate the suggestion to modify the title for greater clarity and precision.
We agree that the proposed wording more accurately reflects the manuscript’s focus on
cellulose–lignin ratio optimization for bio-based packaging materials within the framework of
microRNA-based strategies and avoids possible misinterpretation of the phrase “enhancing
plant biomass quality.” Accordingly, we have revised the title to: “The Role of MicroRNA-
Based Strategies in Optimizing Plant Biomass Composition for Bio-Based Packaging
Materials” (Lines: 2-3).
Comments 2. In paragraph 3, I suggest listing the names of enzymes involved in miRNA
biogenesis in plants. For example: SE (SERRATE, the zinc finger protein) etc.
Response 2. We thank the reviewer for the suggestion. Paragraph 3 has been revised to include
the full names and brief descriptions of the enzymes involved in plant miRNA biogenesis:
DCL1 (DICER-LIKE 1, RNase III endonuclease), SE (SERRATE, zinc finger protein), HYL1
(HYPONASTIC LEAVES 1, double-stranded RNA-binding protein), HEN1 (HUA
ENHANCER 1, methyltransferase), HST1 (HASTY, exportin-5 ortholog), and AGO1
(ARGONAUTE 1, RNA-induced silencing complex protein), (Lines: 119-128).
Comments 3. Please rephrase the sentence (row 117-118):....by RNA polymerase II into
primary miRNAs (pri-miRNAs) which fold into a stem-loop structure.
Response 3. We thank the reviewer for the comment. The sentence has been rephrased for
clarity as follows: In plants, miRNA biogenesis is a multi-step process that begins with the
transcription of miRNA genes by RNA polymerase II into primary miRNAs (pri-miRNAs),
which fold into stem–loop structures [29] (New Lines: 120-121).
Comments 4. Please, include a scheme of how microRNAs regulate genes involved in the
regulation of cell wall formation. How are the individual genes of this process regulated by
microRNA?
Response 4. We have added a schematic diagram (Figure 2) illustrating the regulation of genes
involved in plant cell wall formation by specific microRNAs (Lines: 160-161).
Comments 5. For Paragraph 5.2, please summarise the advantages and disadvantages of each
approach in the context of the review topic, presenting this information in table form.
Response 5. We thank the reviewer for the valuable suggestion. Paragraph 5.2 has been revised
to include a new table (Table 3) that summarises the main advantages and disadvantages of
each biotechnological approach discussed, in the context of optimising cellulose–lignin ratios
for bio-based packaging materials (Lines: 352-354).
Comments 6. It is beneficial that the authors consider the potential risks and disadvantages of
individual approaches, addressing and arguing them specifically (e.g. lines 404–439).
Response 6. We thank the reviewer for this important comment. In addition to the general
discussion provided, we have now expanded Table 3 to include the potential risks and
disadvantages specific to each biotechnological approach. For each method (transgenic
overexpression, target mimicry/STTM, artificial miRNAs, CRISPR/Cas-based genome
editing, and integration with breeding programmes).
Comments 7. For Paragraph 7, please summarize the challenges for miRNA-based approaches
in the context of the review and poin out the potential risks in relation to: modified cell wall
composition versus balanced plant.
Response 7. We thank the reviewer for the suggestion. Paragraph 7 has been revised to
summarise key challenges for miRNA-based approaches and highlight potential risks of
modified cell wall composition versus balanced plant physiology (Lines: 486-492).

Reviewer 2 Report
Comments and Suggestions for Authors
Comments to Authors:
The manuscript titled “The Role of microRNAs in Enhancing Plant Biomass Quality 2 and Productivity” reviews the role of plant microRNAs (miRNAs) in improving plant biomass quality and productivity, with emphasis on cellulose–lignin composition relevant for bio-based packaging. It explains the main biomass components (cellulose, hemicellulose, lignin), outlines miRNA biogenesis and their regulatory roles in growth, stress response, and cell wall formation, and then discusses specific miRNAs such as miR156, miR319, miR397, and miR408 that influence lignin content, cellulose yield, and biomass accumulation. The paper also summarizes biotechnological approaches (target mimicry, artificial miRNAs, STTM, CRISPR) and highlights prospects and challenges in applying miRNA engineering for sustainable packaging feedstocks.
The review is well structured and covers the topic broadly and provides a clear description of miRNA families and their roles. Tables and figures make the content accessible.
However, following points to be addressed before publication.
- The title does not fully reflect the “bio-based packaging” theme. Moreover, Section 6 (prospects of plant miRNA research in bio-based packaging production) lacks substantive discussion of packaging industry applications. Concrete industrial examples should be added.
- The link between miRNA regulation and packaging material production should be emphasized. For example, Table 1 could be expanded to include industrial relevance.
- Several sections are overly descriptive. The manuscript would benefit from more concise writing and stronger critical analysis.
- Section 5.1 mentions advanced web-based platforms for predicting miRNA–mRNA interactions. A new table summarizing these tools, with URLs, would greatly improve utility for readers.
- Several gene names (e.g., DCL1, SE, HYL1) are introduced without explanation. At minimum, their full names and brief functional descriptions should be provided to ensure clarity for readers unfamiliar with these abbreviations.
Author Response
Reviewer 2
Comments 1. The title does not fully reflect the “bio-based packaging” theme. Moreover, Section 6 (prospects of plant miRNA research in bio-based packaging production) lacks substantive discussion of packaging industry applications. Concrete industrial examples should be added.
Response 1. We thank the reviewer for this valuable comment. In accordance with the first reviewer’s suggestion, the title has been revised to “The Role of MicroRNA-Based Strategies in Optimizing Plant Biomass Composition for Bio-Based Packaging Materials” to better reflect the bio-based packaging theme. Furthermore, Section 6 has been substantially expanded to include concrete industrial examples of packaging applications (Lines 2-3, 366-389,414-423).
Comments 2. The link between miRNA regulation and packaging material production should be emphasized. For example, Table 1 could be expanded to include industrial relevance.
Response 2. We thank the reviewer for this valuable comment. The link between miRNA regulation and packaging material production has been revised emphasized in the updated text to ensure clearer connection.
Comments 3. Several sections are overly descriptive. The manuscript would benefit from more concise writing and stronger critical analysis.
Response 3. We thank the reviewer for this valuable comment. The manuscript has been revised to be more concise, with reduced descriptive content and enhanced critical analysis.
Comments 4. Section 5.1 mentions advanced web-based platforms for predicting miRNA–mRNA interactions. A new table summarizing these tools, with URLs, would greatly improve utility for readers.
Response 4. We thank the reviewer for this valuable suggestion. In Section 5.1, we have added a new table summarizing the in silico tools for predicting plant miRNA–mRNA interactions, including their main functions and URLs, to improve clarity and utility for readers (Lines 259-260).
Comments 5. Several gene names (e.g., DCL1, SE, HYL1) are introduced without explanation. At minimum, their full names and brief functional descriptions should be provided to ensure clarity for readers unfamiliar with these abbreviations .
Response 5. We thank the reviewer for pointing this out. In the revised manuscript, we have provided the full names and brief functional descriptions for all gene abbreviations (Lines 121-127, 532-533).

Round 2
Reviewer 2 Report
Comments and Suggestions for Authors
Authors have addressed the comments thoughtfully and made substantial improvements to the manuscript. Overall, these revisions have significantly enhanced the manuscript’s quality and make it suitable for publication.
Thanks for your efforts.